# How Does the Male Penisfilum Enter the Female Copulatory Pore in Hangingflies?

**DOI:** 10.3390/insects11020123

**Published:** 2020-02-14

**Authors:** Zheng Wei, Xin Tong, Bao-Zhen Hua

**Affiliations:** Key Laboratory of Plant Protection Resources and Pest Management, Ministry of Education, College of Plant Protection, Northwest A&F University, Yangling 712100, China; weizheng@nwafu.edu.cn (Z.W.); tong.xin@nwafu.edu.cn (X.T.)

**Keywords:** abdomen, twist, swelling, genital morphology, copulatory mechanism

## Abstract

Hangingflies are characterized by the interesting nuptial feeding behavior and unusual belly-to-belly hanging mating position. However, the mating behavior and the copulatory mechanism remain poorly known for Bittacidae, especially how the elongated male penisfilum enters the copulatory pore of the female. In this study, the mating behavior and copulatory mechanism of *Terrobittacus implicatus* (Huang and Hua, 2006) were investigated to reveal the functional morphology of hangingfly genitalia. The results show that the male provides a prey as a nuptial gift to the female and twists his abdomen about 180° to form a belly-to-belly hanging mating position. During the penisfilum-entering process, the male epandrial lobes clamp the female subgenital plate with the aid of the female abdomen swelling. Then the male locates the female copulatory pore through his upper branch of the proctiger and inserts his penisfilum into the female spermathecal duct in cooperation with the short setae on the groove of the proctiger. The female subgenital plate where the epandrial lobes clamp is strongly sclerotized and melanized. The copulatory mechanism of *Terrobittacus* is briefly discussed.

## 1. Introduction

Most animals use internal fertilization to directly transfer sperm during copulation [1]. Therefore, firm coupling of the genitalia between the two sexes is critical for the success of mating [2,3]. For insects, most males evolved behavioral and morphological strategies to grasp and control their mates, including modified genital structures, diverse mating positions and nuptial feeding behaviors to obtain advantage in sexual conflict [4,5,6,7].

Mecoptera are often of interest to entomologists because of their diverse grasping devices and mating positions [8]. The males of Boreidae use their tong-like reduced wings to secure the female to form a female-above position [7,9]. Most male Panorpidae use the notal organ on the posterior margin of the third abdominal tergum and the anal horn(s), if present, on the posterior margin of the sixth abdominal tergum to clamp the female to sustain a V-shaped mating position [10,11,12,13]. In particular, the males of *Furcatopanorpa* (Panorpidae) and *Chorista* (Choristidae) lacking a notal organ adopt an O-shaped mating position [8,14]. The males of Panorpodidae, which lack a notal organ, use their modified genitalia to grasp the female and form an end-to-end mating position [15]. However, detail about the mating behavior and functional morphology of Mecoptera is still poorly studied in many groups, including Bittacidae.

Bittacidae are characterized by an unusual belly-to-belly hanging mating position, which is often accompanied by the twisting of the male abdomen [16,17]. In general, male hangingflies provide a flying insect prey as a nuptial gift to the female prior to copulation, then provisionally twist their abdomen by 180° clockwise or counterclockwise to form a belly-to-belly hanging mating position [16,17,18,19,20,21,22,23]. To adapt to this unique mating position, the male genitalia have undergone marked modifications for their extremely elongated penisfilum, well-developed epandrial lobes (tergum IX), greatly reduced gonostyli and the sophisticated proctiger [17]. During copulation, the modified epandrial lobes are used to grasp the female abdomen, mainly the subgenital plate [17,18]. Although the mating behavior and copulatory mechanism have been studied for several species in Bittacidae [17,18,24,25,26,27,28], it is still unknown how the greatly elongated male penisfilum precisely finds and enters the copulatory pore of the female and what function the uniquely-structured proctiger serves during copulation.

In this study, the mating behavior and copulatory mechanism of hangingfly *Terrobittacus implicatus* (Huang and Hua, 2006) were observed in detail to clarify the functional morphology of Bittacidae genitalia.

## 2. Materials and Methods

### 2.1. Insect Collection

About 180 adults of *Terrobittacus implicatus* (Huang and Hua, 2006) were collected at Liping Forest Park (32°47′ N, 106°40′ E, 1500–2300 m a.s.l.), Shaanxi Province of Central China from late July to early August from 2017 to 2019.

### 2.2. Insect Rearing

The male and female adults were all reared together in two gauze cages (40 cm × 40 cm × 60 cm) in the laboratory under natural conditions. Mating behavior was observed from 14–30 July, 2019. Absorbent cotton containing water was replaced each day at the bottom of cages. Twigs with leaves were placed in the cages to simulate shaded microenvironments for the adults and were replaced every three days [17]. Live flies (blow flies and house flies), crane flies and other flying insects captured from the fields were supplied as food items. Each individual was supplied three prey items per day. The cage was cleaned regularly every day.

### 2.3. Mating Behavior Observation

Ten males and ten females were reared together in a cage and mated with each other multiple times to observe their mating behaviors (about 110 pairs), including mating processes (about 80 pairs) and copulation duration (23 pairs). Photographs were taken with a Nikon D200 digital camera.

### 2.4. Light and Scanning Electron Microscopy

Twenty males and ten females were fixed in Carnoy’s solution for morphological studies.

The male and female genitalia were dissected under a Nikon SMZ1500 microscope (Nikon, Tokyo, Japan). Photographs were taken with a QImaging Retiga-2000R Fast 1394 digital camera (QImaging, Surrey, Canada) attached on the microscope and were stacked with Syncroscopy Auto-Montage software (Syncroscopy, Cambridge, UK).

For scanning electron microscopy (SEM), the dissected genital structures were cleaned ultrasonically for 70 s before being dehydrated in a graded ethanol series, freeze-dried for 3 h, coated with gold in a sputter coater and observed in a Hitachi S-3400N scanning electron microscope (Hitachi, Tokyo, Japan) at 15 kV.

## 3. Results

### 3.1. Genital Structures of the Male

The male genitalia of *T. implicatus* consist of a pair of dorsal epandrial lobes, a proctiger, a central aedeagal complex and a pair of ventral gonopods (Figure 1A,B).

The epandrial lobes are paired claspers modified from tergite IX, roughly subtriangular in lateral aspect (Figure 1A). The small epandrial lobes coalesce basally and are shorter than half the length of gonocoxites, with numerous long hairs (Figure 1A,E). Each epandrial lobe bears a patch of dense black spines on the apical inner margin (Figure 1B,E). The spines are conical with radial grooves (Figure 1H).

The proctiger consists of an upper branch and a lower branch (Figure 1A,B). The upper branch is strongly sclerotized dorsolaterally and protrudes from between the bases of the epandrial lobes, with the apex curved caudoventrally into a hook (Figure 1A). The upper branch has a deep caudal groove, which has a row of dense short setae (Figure 1F). The base of the upper branch is outwardly expanded (Figure 1B,D). The lower branch is considerably shorter and broader basally (Figure 1A). The apex of the lower branch is slender, with numerous setae in the dorsal groove (Figure 1G). The anus is situated between the upper and lower branches and bears dense short hairs (Figure 1D). The paired cerci are shorter than the upper branch of the proctiger (Figure 1A).

The aedeagal complex consists of a penisfilum, a pair of aedeagal lobes, a phallobase and an epiphallus (Figure 1C). The penisfilum is greatly prolonged, slender toward the apex and coiled into a loop (Figure 1A,C). The paired aedeagal lobes are acute and are situated on the basal sides of the aedeagus (Figure 1B,C).

The gonopods consist of a pair of basal gonocoxites and a pair of distal gonostyli (Figure 1A). The gonocoxites are posteroventrally rounded with a V-shaped pale medial area (Figure 1B). The greatly reduced gonostyli are clothed with dense setae (Figure 1B).

### 3.2. Genital Structures of the Female

The main structure of the female genitalia is the strongly sclerotized subgenital plate (Figure 2A,B), which is composed by two halves and is black along the midventral line in the basal half (Figure 2A,C). Trichoid sensilla are near the opening of the genital chamber (Figure 2D). The copulatory pore (or the orifice of the spermathecal duct) is located at the posterior end of the common oviduct in the genital chamber. The spermatheca is spheroid with an elongate spermathecal duct (Figure 2E,F). The fine spermathecal duct coils elliptically and is located beside the spermatheca (Figure 2F). The spermathecal duct leaves the coils and gradually becomes thicker toward the posterior end of the common oviduct (Figure 2F).

### 3.3. Mating Behavior

The mating process can be divided into three phases: calling (about 90% of total mating pairs; some males cannot attract a female), pairing (about 80% of total mating pairs; some females are unsatisfied with the gift) and copulating phases (about 80% of total mating pairs; some males cannot insert their penisfilum into the spermathecal duct of the female). When a male captured a prey, he first fed upon the prey for a short palatability evaluation and then made a short flight to find a suitable mating place (Figure 3A). Once having a suitable prey, the male suspended the prey by his hindlegs and bent his abdomen ventrally into a J-shape. The male everted his sex pheromone glands between abdominal terga VI and VII and terga VII and VIII and raised his wings by approximately 30°, entering into the calling phase (Figure 3B).

During the calling phase, the male released sex pheromones (by the eversible sex pheromone glands) to attract a female. When a female approached, the male gave her the prey, initiating the pairing phase (Figure 3C).

During the pairing phase, the female fed on the prey to assess the size and palatability of the nuptial gift. If the female was satisfied with the gift, she bent her abdomen dorsally to show the willingness to mate (Figure 3D). The male attempted to reach his genitalia to seek the tip of the female abdomen and entered into the copulating phase.

During the copulating phase, the male twisted his abdomen up to approximately 180° and formed a belly-to-belly hanging mating position (Figure 3F).

At the end of copulation, the female twisted her abdomen to break the male control. Then the male disengaged his epandrial lobes and turned his abdomen back to the normal position. The copulation duration lasted about 30 min (mean ± SD = 22.78 ± 13.59 min, *n* = 23).

### 3.4. Entering of the Male Penisfilum

The entering of the male penisfilum can be subdivided into two steps: clamping of the male epandrial lobes and platforming of the male proctiger.

#### 3.4.1. Clamping of the Male Epandrial Lobes

When the female showed her willingness to mate, the male spread his epandrial lobes widely and bent his gonopods ventrally (Figure 3D). Then the male used his abdomen to twine the tip of her abdomen either clockwise or counterclockwise (Figure 3E). The male made a short pause at the tip of the female tergum, getting a positioning (Figure 3E). If the positioning was unsuccessful, the male departed from the female abdomen and tried again. If the positioning was successful, the male continued to twist his abdomen along the female abdomen (Figure 4A).

When the male terminalia touched the tip of the female abdomen, the female abdomen expanded from segments III to VII (Figure 4B,C). Then the male epandrial lobes moved along the route formed by segments VII and VIII of the female (Figure 4B,C). During this process, the epandrial lobes of the male closely attached and firmly clamped the female subgenital plate (Figure 4). The dense black spines on the inner margin of the epandrial lobes touched the melanized area of the subgenital plate (Figure 5D). Then the female abdomen retracted and the tegument of abdominal segment VII covered the tip of the epandrial lobes (Figure 4D).

#### 3.4.2. Assistance of the Male Proctiger

After grasping the female, the male attempted to locate the female copulatory pore through the upper branch of his proctiger (Figure 5A). The male moved his gonopods ventrally, thus enabling the penisfilum to connect the lower branch of his proctiger. The male adjusted his penisfilum into the groove of the lower branch (Figure 6C).

The male bent his genitalia ventrally again. The extremely elongated penisfilum was stretched along the lower branch of the proctiger (Figure 4D and Figure 5B). The female bent her abdomen ventrally to fit the insertion of the penisfilum (Figure 3F). The penisfilum entered into the female genital chamber along the caudal groove of the proctiger (Figure 6A). The dense short setae on the groove of the proctiger helped the penisfilum insert into the female spermathecal duct (Figure 2E and Figure 6B). The male gradually inserted his penisfilum into the spermathecal duct of the female with the aid of the proctiger and gonopods (Figure 5B,C). When the penisfilum was fully inserted, the male genitalia fit those of the female closely to transfer sperm (Figure 5D).

## 4. Discussion

In this study, the mating behavior and copulatory mechanism of *Terrobittacus* were described. The male and female maintain a belly-to-belly hanging mating position during mating. To assume the unique mating position, the male grasps the female with the help of epandrial lobes. Then the male inserts his elongated penisfilum into the female spermathecal duct with the assistance of the proctiger. It is likely that this is the first time that the detail of the elongated male penisfilum entering into the female copulatory pore in Bittacidae is reported.

The genital and nongenital grasping devices usually maintain the firm coupling of genitalia between the two sexes in sexually reproductive insects [1,11,14,15,29,30,31], such as the male cerci in damselflies [32], the specialized abdominal apparatus in water striders [33,34,35,36] and the male epandrial lobes in hangingflies [37,38]. The developed epandrial lobes of *Bittacus planus* can grasp part of the female abdomen to maintain the firm coupling of the two sexes [17,37]. However, the small epandrial lobes of *T. implicatus* clamp a very small area on the female abdomen. Therefore, the epandrial lobes adaptively bear a patch of black spines, which may help the male to firmly clamp the female abdomen by increasing friction. Due to the stimulation of the black spines and the strong clamping from the male epandrial lobes, the subgenital plate of the female is strongly sclerotized and melanized. It is implied that sexually antagonistic coevolution may occur in the epandrial lobes of male and the subgenital plate of the female in *T. implicatus*.

The degree of sclerotization of the female subgenital plate is likely related to the size of the male epandrial lobes. The epandrial lobes of *B. planus* are roughly quadrangular in lateral aspect and furnished with black spines sparsely distributed along the inner posterior edge. The subgenital plate of *B. planus* is heavily sclerotized but not melanized [17]. However, the small epandrial lobes of *T. implicatus* adaptively bear a patch of black spines and the subgenital plate is strongly sclerotized and melanized. We speculated that if the epandrial lobes are small, they may bear more spines to increase friction. The female abdomen may receive more force on a smaller area so that the degree of sclerotization of the female subgenital plate is stronger.

The proctigers of Bittacidae show a large diversity among different genera [37,38,39,40]. The proctiger of *T. implicatus* is more developed and more strongly sclerotized than that of most other hangingflies. In the copulating phase, the short setae on the groove of the proctiger assist the elongated penisfilum into the female spermathecal duct. The bases of the upper branch of the proctiger in *T. implicatus* are outward. During copulation, the outward upper branch of the proctiger enters the female genital chamber. Compared with the slender proctiger of *B. planus* [17], the outward proctiger of *T. implicatus* may be more difficult to remove from the female genital chamber. We considered the developed proctiger is likely to help the male maintain the firm coupling of the two sexes.

The physical stimulation of male genitalia to the female abdomen is likely to be related to triggering copulation and/or increasing ovulation [41,42,43]. In scorpionflies, males open the female genital chamber with the aid of physical stimulation from the male gonostyli, triggering copulation [11,13,15]. In stick insects, the trichoid sensilla of the female abdomen are able to feel the male grasping, then the female opens her subgenital plate [44]. Male hangingflies cannot open the female genital chamber by their gonostyli since the gonostyli are greatly reduced. However, the trichoid sensilla on the female subgenital plate of *T. implicatus* are located near the opening of the genital chamber. Given that the function of the trichoid sensilla of the female subgenital plate of *T. implicatus*, as in stick insects, may detect the mechanical stimulation from the male, then the female opens her genital chamber, thus triggering the mating behavior.

Male hangingflies twist their abdomen by 180° to maintain a belly-to-belly hanging mating position [8]. In fact, many insects twist their abdomen to assume diverse mating positions during copulation, such as Diptera [45,46,47,48,49,50,51]. For many dipteran species that twist the abdomen during copulation, the male genitalia rotate to a certain angle during the pupal and adult stages, temporarily or permanently [52,53,54]. However, the abdomen twists distinctly in hangingflies during the mating process. The males of *T. implicatus* with small epandrial lobes twist their abdomen and have great difficulty in clamping the female subgenital plate accurately. Therefore, when the male genitalia touch the female abdomen, the female significantly swells her abdomen from segments III to VII, which can cause a distinct difference in the diameters between segments VII and VIII. The male twists his abdomen using his epandrial lobes to move along the route formed by abdominal segments VII and VIII of the female to accurately clamp the female subgenital plate. The female may cooperate with the male to accomplish copulation. Female hangingflies choose males with large prey items as gifts for mating [28]. This female behavior constitutes a cryptic female choice trait. Females not only can directly gain considerable nutritional benefits from nuptial gifts but also indirectly gain good genetic benefits for their offspring [3]. Therefore, we suggest that when females choose the suitable males, they can cooperate with males to connect their copulatory pore.

## 5. Conclusions

During copulation, the female of *T. implicatus* swells her abdomen to help the male touch her subgenital plate. The setae on the groove of the proctiger assist the elongated male penisfilum with entering the female abdomen. The small epandrial lobes of the male bear a patch of black spines to firmly clamp the female abdomen by increasing friction. The subgenital plate of the female is strongly sclerotized and melanized because of clamping from the male epandrial lobes.

## Figures and Tables

**Figure 1 insects-11-00123-f001:**
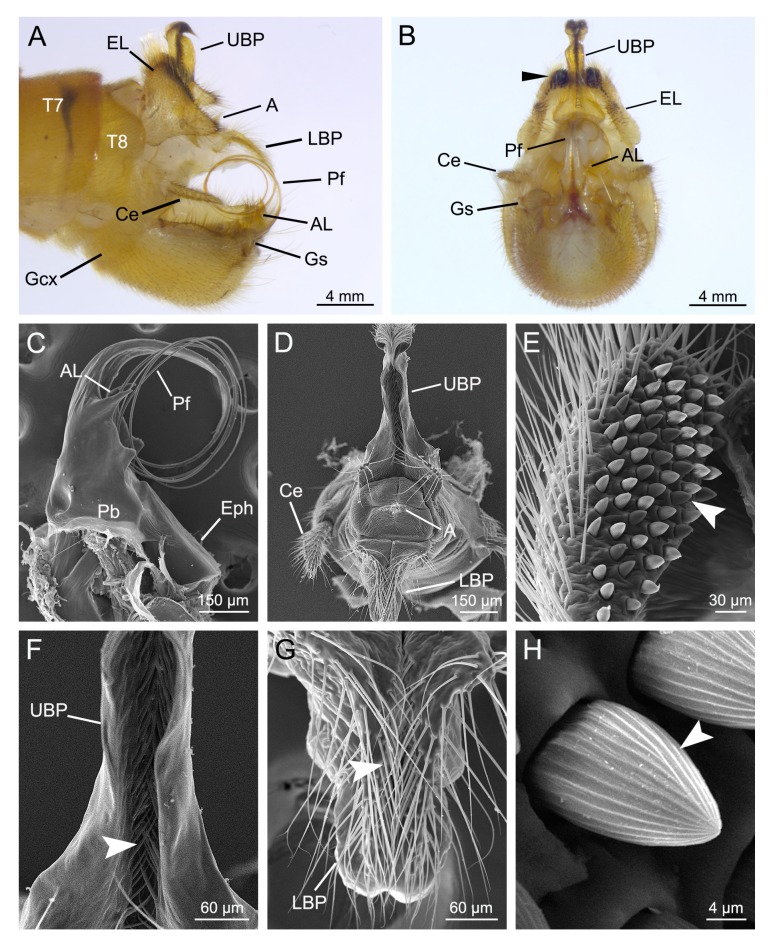
Male genitalia of *Terrobittacus implicatus*. (**A**) and (**B**) are auto-montage micrographs. (**C**–**H**) are scanning electron microscopy (SEM) micrographs. (**A**) Male genitalia, lateral view. (**B**) Male genitalia, caudal view. Arrowhead shows spines on the epandrial lobes. (**C**) Aedeagal complex, lateral view. (**D**) Proctiger, caudal view. (**E**) Epandrial lobe, caudal view. Arrowhead shows spines. (**F**) Magnification of the upper branch of proctiger to show the setae (arrowhead). (**G**) Magnification of the lower branch of proctiger to show the setae (arrowhead). (**H**) Magnification of the spine on the epandrial lobe. Arrowhead shows the radial grooves on the spine. Abbreviations: A, anus; AL, aedeagal lobe; Ce, cercus; EL, epandrial lobe; Eph, epiphallus; Gcx, gonocoxite; Gs, gonostylus; LBP, lower branch of proctiger; Pb, phallobase; Pf, penisfilum; UBP, upper branch of proctiger; T7–T8, terga VII–VIII.

**Figure 2 insects-11-00123-f002:**
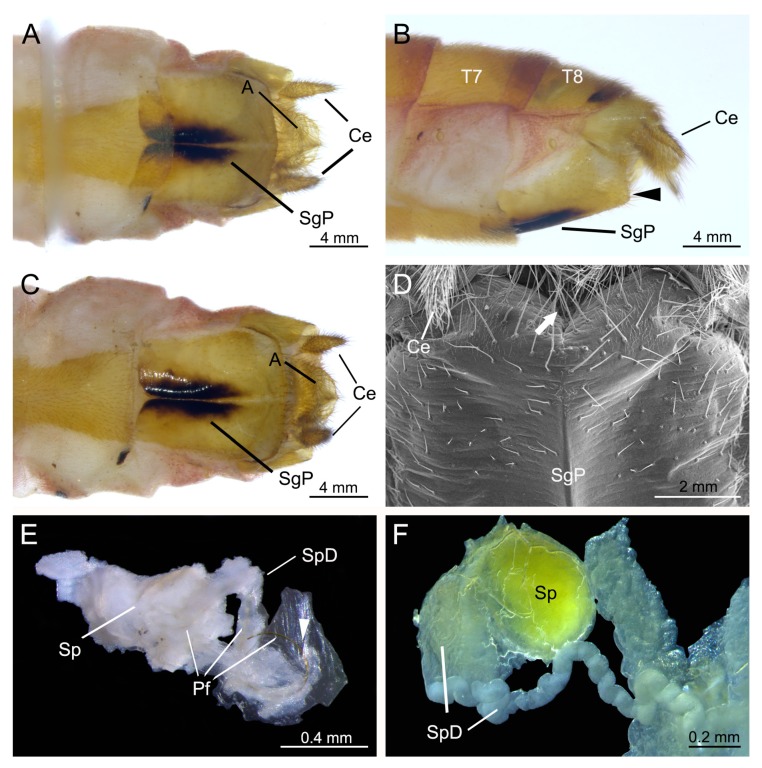
Female genitalia of *Terrobittacus implicatus*. (**A**–**C**) and (**E**,**F**) are auto-montage micrographs. (**E**,**F**) are dissected parts. (**D**) is an SEM micrograph. (**A**) Terminal end of abdomen in ventral view, natural condition of sternum VII. (**B**) Terminal end of abdomen in lateral view. Arrowhead shows the opening of the genital chamber. (**C**) Terminal end of female abdomen in ventral view, with sternum VIII pulled out. (**D**) The opening of genital chamber, caudo-ventral view. Arrowhead shows the opening of the genital chamber. (**E**) Part of the female reproductive system, showing the elongate spermathecal duct with male penisfilum inside. Arrowhead shows the orifice of common oviduct. (**F**) Magnification of the spermatheca and elongate spermathecal duct. Abbreviations: A, anus; Ce, cercus; Pf: penisfilum; SgP, subgenital plate; Sp: spermatheca; SpD: spermathecal duct; T7–T8, terga VII–VIII.

**Figure 3 insects-11-00123-f003:**
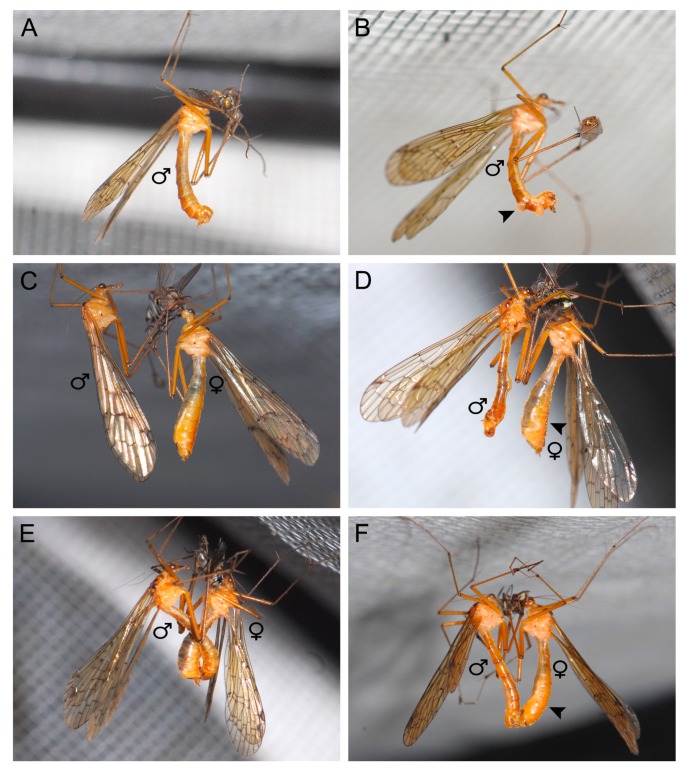
Mating process of *Terrobittacus implicatus*. (**A**,**B**) Calling phase. (**A**) A male feeds upon the prey for a short while to taste its palatability. (**B**) A male everts his sex pheromone glands (arrowhead) between terga VI and VII and terga VII and VIII to release sex pheromone to attract females nearby and raises his wings by 30°. (**C**,**D**) Pairing phase. (**C**) The male presents the prey as a nuptial gift to the female, and the female accepts the nuptial gift and begins to assess its size and palatability. (**D**) The male opens his epandrial lobes widely and his genitalia inflate, and the female shows her willingness to mate by bending her abdomen (arrowhead) dorsally. (**E**,**F**) Copulating phase. (**E**) The male uses his abdomen to twine the tip of the female abdomen, and the female feeds on the gift. (**F**) The male provisionally twists his abdomen along his longitudinal axis up to 180° to adapt to the belly-to-belly mating with the female in hanging position. Arrowhead shows the female bending her abdomen ventrally.

**Figure 4 insects-11-00123-f004:**
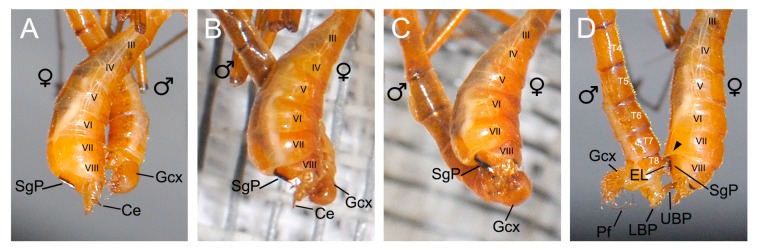
The dorsal end (female segments III-VIII) of copulating *Terrobittacus implicatus*, showing the swelling process of the female abdomen. (**A**) The male makes a short pause at the tip of the female tergum, getting a positioning. (**B**) The female abdomen begins to swell from segments III to VII. (**C**) The epandrial lobes of the male move along the route formed by the abdominal segments VII and VIII of the female. (**D**) The female abdomen restores and the tegument (arrowhead) of the abdominal sternum VII covers the tip of the male epandrial lobes. Abbreviations: Ce, cercus; EL, epandrial lobe; Gcx, gonocoxite; LBP, lower branch of proctiger; Pf, penisfilum; SgP, subgenital plate; UBP, upper branch of proctiger; T4–T8, terga IV–VIII.

**Figure 5 insects-11-00123-f005:**
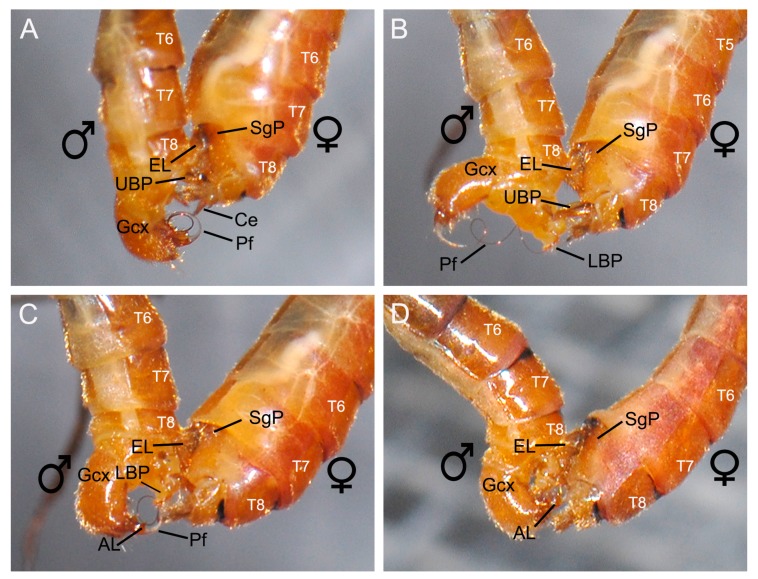
The dorsal end (female segments III-VIII) of copulating *Terrobittacus implicatus*, showing the gradual insertion process of the male penisfilum. (**A**) The male firmly holds the subgenital plate of the female by his epandrial lobes and the upper branch of the proctiger extends into the female genital chamber. (**B**) The elongated penisfilum is stretched by the lower branch of the proctiger and the proctiger assists the entering of the male penisfilum into the female. (**C**) The partly inserted penisfilum. (**D**) The male penisfilum is fully inserted into the female. Abbreviations: AL, aedeagal lobe; Ce, cercus; EL, epandrial lobe; Gcx, gonocoxite; LBP, lower branch of proctiger; Pf, penisfilum; SgP, subgenital plate; UBP, upper branch of proctiger; T5–T8, terga V–VIII.

**Figure 6 insects-11-00123-f006:**
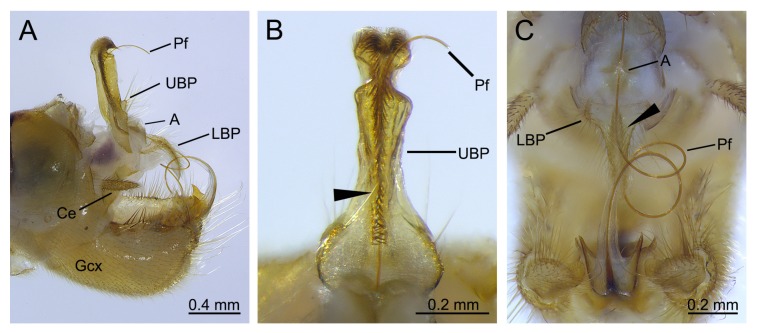
The penisfilum during *Terrobittacus implicatus* copulation, showing the pattern that the proctiger assists the penisfilum. (**A**) The penisfilum is meshed in the caudal groove of the proctiger. (**B**) The penisfilum is meshed in the groove of the upper branch of the proctiger. Arrowhead shows the dense short setae on the caudal groove. (**C**) The penisfilum is meshed in the groove of the lower branch of the proctiger. Arrowhead shows the dense short setae on the caudal groove. Abbreviations: A, anus; Ce, cercus; Gcx, gonocoxite; LBP, lower branch of proctiger; Pf, penisfilum; UBP, upper branch of proctiger.

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
