# Peer review of "How Does the Male Penisfilum Enter the Female Copulatory Pore in Hangingflies?"

_insects, 2020, doi:10.3390/insects11020123_

Round 1
Reviewer 1 Report
General comments
The manuscript “How the male elongated penisfilum enters into the female copulatory pore for hangingflies?” investigates the mating behaviour and the male and female genitalia functional morphology of the hanging fly Terrobittacus implicatus (Mecoptera, Bittacidae).
The genital structures of males and females were examined by means of optical and scanning electron microscopy. The Authors describe the mating phases in T. implicatus and for the first time, through their microscopic analyses, they show the male and female genital morphological adaptations, thus clarifying the peculiar copulatory process in this species.
This study contributes towards building a better understanding of the reproductive behaviour of a poorly studied group. The results will be of wide interest for the readers of Insects and are worthwhile to be published. I do have some comments and suggestions for the improvement of the manuscript.
If I understand correctly, the Authors analysed the mating behaviour of 23 mating pairs. It would be interesting and useful to have more information. For instance, it is not unlikely that males sometimes struggle to end copulation. In this respect, did they Authors record the behaviour of 23 mating pairs in total (and they were all successful matings), or did they record the behaviour of more mating pairs and discard the unsuccessful matings?
The Authors mention the possibility of sexually antagonistc coevolution (L234-236), given the presence of the strongly sclerotized and melanized subgenital plate in the females. I am curious to know whether they observed any antagonistic interactions between sexes in this hanging fly. It seems that T. implicatus males cannot copulate without female cooperation, ruling out male coercive behaviour. Might this suggest a victory of coevolutionary arms race by the female sex? Moreover, the high diversity of the proctiger within Bittacidae (L237) might indicate a transition to mating systems with differing levels of sexual conflict.
In general, the great accessibility to the offered food gift makes it less likely the occurrence of arms race between males and females. Could perhaps food availability, which is high in lab condition, influence male mating behaviour?
Perhaps the Authors might want to expand a bit more on these interesting issues.
Finally, females choosing males with larger nuptial gifts exert a pre-copulatory choice, rather than a cryptic choice (L265). Are females of this hanging fly polyandrous? Can they control copulation duration? If females are able to manipulate the number of sperm transferred, then a pre-copulatory preference for a male with a larger nuptial gift can be reinforced by a post-copulatory cryptic female choice.
Minor comments:
The Authors might want to reconsider the title. I suggest the following:“How do the male penisfilum enter the female copulatory pore in hangingflies?” L45: “…male genitalia underwent…” L57-68: How many insects were collected in total? L87: Please, replace “The same” by “Male genitalia” L88 and throughout the whole text: Please, replace “Arrow” by “Arrow head” L147 and throughout the whole text: Please, replace “mating desire” by “willingness to mate” L161: Do bigger nuptial gifts result in copulations of longer duration?Author Response
Please see the attachment.

Reviewer 2 Report
This manuscript provides an observation-based description of the Terrobittacus implicatus mating process. The description covers fine details of how male and female genitalia interact during copulation. Overall the manuscript is fairly well written, it is logical and easy to follow. Excellent figures are used to demonstrate each stage of how the mating process has been described, the figures are detailed and clear. These are the strongest point of the manuscript and provide detail at different levels from the perspective of the whole organism to close ups of specific anatomy. I would love to see these figures published in the literature.
I personally place a high value on natural history and want to see this work published. However, as an observation piece, I am not sure this work fits with the scope and aims of the journal Insects who’s “aim is to encourage scientists to publish their experimental and theoretical results in as much detail as possible”. This is a superb description of the Terrobittacus implicatus mating process. However, this manuscript does not provide any results. There seems like there is the potential to add results- if you quantify each stage of the mating process. It seems like you would have the information to be able to do this. Currently, what you have provided is a solid description, which makes an excellent catalyst to draw hypothesis from, that you can then test to get results. This manuscript is suitable for submission to a journal where natural history is in the scope of their publications.
Before this work is published. More thought needs to go into the writing and concepts you are trying to address, there are many instances that require further explanation. I have offered some suggestions on where improvements can be made. My suggestions need to be applied throughout the paper- not only where I have made them.
Lines 15-16: “The results show……..twists his abdomen 180°”. The term “results” suggests you have measured something. This manuscript does not include any measurements at all. If you did measure the angle the male twists his abdomen, you have not explained that this was done. You have observed, described and documented. However, you have not presented results.
Line 26: “Most internally fertilizing animals transfer sperm directly through copulation [1–3].”
This is an odd sentence. Could be: Most animals use internal fertilization to directly transfer sperm during copulation.
Then the references you have used here do not back up this statement. The articles 1 & 2 do not even mention internal fertilization.
Line 31: “attracts” is rather emotive. Appropriate alternatives- is of interest to, relevant, regularly studied.
Line 45: remove “underwent”
Line 54: replace “investigated” with observed.
Line 58: State how many adults were collected. If this work was to be repeated it’s important for the reader to know how many to begin with. I know from my own insect mating experiments. I end up collecting a lot more than I can observe (many die along the way). If you have a 100% success rate, that is very impressive.
Line 62: Were males and females reared separately or together in two different cages? This is currently unclear.
Line 63. Please include the observation period – the dates observations were carried out between.
Line 70: When observing mating behavior were all 10 males and females together in the same cage? Or did you put pairs in separate cages? This needs to be explained.
Line 71: How did you take the photos? Did you simply point and shoot aiming to be parallel with an anatomy? Use a stand? The photo’s in your figures are nicely uniform, it’s beneficial to include how this was achieved.
Line 73: Are these individuals the same individuals that were used in the mating experiments or different?
Line 87: Figure 1: Change “ the same“ to male genitalia. This figure legend needs to state which images are auto-montage and which are SEM.
Line 125: Figure 2: This figure legend needs to state which images are auto-montage, which are SEM and state that E and F are dissected parts.
Lines 134-161: The mating behavior process is written as though this observation is only based on one pair. Did the exact same process happen for all ten of your mating pairs?
For each phase it would be beneficial to include how many times this phase was observed.
Line 135: “when a male captured a prey” – this means you must have included prey items in their observation cages. You need to state what they were and how many in your methods.
Line 138: What does regularly mean? Did this occur for an extended period of time, was it fast, slow, observable up to 10 times, 100’s of times?
Line 151: “the male released sex pheromones to attract a female”. How do you know this? This is what you are presuming, but you can’t see the sex pheromones. Has other work shown that a particular action is associated with pheromone release? If so, you need to explain this. If not, experiments need to be done to show that this is the case.
Line 154: “If the female was satisfied with the gift”. This is why it’s good to be quantifying how many times each stage happened. Here clearly, some rejected while others accepted. How many times did each situation occur? Including a table of the frequency of each stage would be a great addition.
Line 161: “n=23)” Only now do you give an indication of how many observations were made. You observed 23 successful copulations, from how many observed pairings? In your methods, you state 10 males and 10 females were used for behavioral observations. Does this mean you paired up individuals’ multiple times? You need to state this in your methods if this is the case.
Line 166: How does the female “show her mating desire”
Line 173: Figure 4: “Copulating phase of Terrobittacus implicatus” should be; The dorsal end (female segments III-VIII) of copulating Terrobittacus implicatus.
Line 188: Figure 5 same as above.
Line 201: figure 6: “Copulating phase of Terrobittacus implicatus” should be -The penisfilum during Terrobittacus implicatus copulation.
Lines 231-244: The discussion suggests some extreme implications. The ideas here are relevant. However, they need to explain that this mating description they have provided here sets the base line of the mating process so that future work can hypothesize, manipulate the system and compare to work towards figuring out whether:
“If the epandrial lobes are small, the female abdomen may receive more force on a smaller area so that the degree of sclerotization of female subgenital plate is stronger. It is implied that sexually antagonistic coevolution may occur in the epandrial lobes of male and the subgenital plate of the female in implicatus.” “The developed proctiger is likely to help the male maintain the firm coupling of the two sexes.” “The physical stimulation of male genitalia to the female abdomen is speculated to be related to trigger copulation and/or increase ovulation”Lines 250-251: “as the same of stick insects” this is suggesting the trichoid sensilla of stick insects and hanging flies have the same. Since you are comparing two extremely divergent lineages (members of different orders), you perhaps can’t be this direct with your comparison. It is indeed likely that the trichoid sensilla of Clitarchus stick insects T. and implicatus are analogous, providing the same function. As both undergo prolonged courtship and mating, this is a good comparison to be making. You just need to explain this clearly.
Reviewer 3 Report
This very interesting paper is a great inspiration and the journal should have no hesitation publishing. The text is well written and the images clear. Explanations are logically made and the conclusions drawn are strongly substantiated by the results.
I detect two minor typos that may need attention prior to publication:
Line 208: insert a space between branch and of. It seems to me that the two words currently abutt with no space between them.
Line 220: sentences is awkwardly written. Suggest change to: It is likely that this is the first time that the detail of the male elongated penisfilum entering into the female copulatory pore in Bittacidae is reported.
